# Influence of CO_2_ and Dust on the Survival of Non-Resistant and Multi-Resistant Airborne *E. coli* Strains

**DOI:** 10.3390/antibiotics13060558

**Published:** 2024-06-14

**Authors:** Viktoria Agarwal, Elena Abd El, Silvia Giulia Danelli, Elena Gatta, Dario Massabò, Federico Mazzei, Benedikt Meier, Paolo Prati, Virginia Vernocchi, Jing Wang

**Affiliations:** 1Institute of Environmental Engineering, ETH Zurich, 8983 Zurich, Switzerland; tepperv@ethz.ch (V.A.); meier-benedikt@gmx.de (B.M.); 2Laboratory for Advanced Analytical Technologies, Empa, Swiss Federal Laboratories for Materials Science and Technology, 8600 Dübendorf, Switzerland; 3Dipartimento di Fisica, Università di Genova, Via Dodecaneso 33, 16146 Genoa, Italy; elena.abd@ge.infn.it (E.A.E.); silviagiulia.danelli@pm10-ambiente.it (S.G.D.); elena.gatta@unige.it (E.G.); dario.massabo@ge.infn.it (D.M.); federico.mazzei@ge.infn.it (F.M.); prati@ge.infn.it (P.P.); 4INFN—Sezione di Genova, Via Dodecaneso 33, 16146 Genoa, Italy; virginia.vernocchi@ge.infn.it

**Keywords:** antimicrobial resistance, bio-aerosols, survival rate, environmental factors, atmospheric simulation chamber

## Abstract

The airborne transmission of bacterial pathogens poses a significant challenge to public health, especially with the emergence of antibiotic-resistant strains. This study investigated environmental factors influencing the survival of airborne bacteria, focusing on the effects of different carbon dioxide (CO_2_) and dust concentrations. The experiments were conducted in an atmospheric simulation chamber using the non-resistant wild-type *E. coli* K12 (JM109) and a multi-resistant variant (JM109-pEC958). Different CO_2_ (100 ppm, 800 ppm, 3000 ppm) and dust concentrations (250 µg m^−3^, 500 µg m^−3^, 2000 µg m^−3^) were tested to encompass a wide range of CO_2_ and dust levels. The results revealed that JM109-pEC958 exhibited greater resilience to high CO_2_ and dust concentrations compared to its non-resistant counterpart. At 3000 ppm CO_2_, the survival rate of JM109 was significantly reduced, while the survival rate of JM109-pEC958 remained unaffected. At the dust concentration of 250 µg m^−3^, JM109 exhibited significantly reduced survival, whereas JM109-pEC958 did not. When the dust concentration was increased to 500 and 2000 µg m^−3^, even the JM109-pEC958 experienced substantially reduced survival rates, which were still significantly higher than those of its non-resistant counterpart at these concentrations. These findings suggest that multi-resistant *E. coli* strains possess mechanisms enabling them to endure extreme environmental conditions better than non-resistant strains, potentially involving regulatory genes or efflux pumps. The study underscores the importance of understanding bacterial adaptation strategies to develop effective mitigation approaches against antibiotic-resistant bacteria in atmospheric environments. Overall, this study provides valuable insights into the interplay between environmental stressors and bacterial survival, serving as a foundational step towards elucidating the adaptation mechanisms of multi-resistant bacteria and informing strategies for combating antibiotic resistance in the atmosphere.

## 1. Introduction

The spread of bacterial pathogens through the air presents a significant challenge to public health, particularly with the emergence and proliferation of antibiotic-resistant strains [1]. Understanding the environmental factors that influence the survival of these bacteria in aerosolized form is crucial for devising effective strategies to mitigate their transmission and impact on human health.

Airborne bacterial survival is influenced by a myriad of environmental factors, both abiotic and biotic [2,3,4,5]. Among these, temperature, relative humidity (RH), and exposure to UV radiation have been extensively studied [4,6,7,8]. Temperature and RH play pivotal roles in bacterial viability, with warm temperatures and moderate RH levels generally favoring bacterial survival and growth. Conversely, extreme temperatures or RH levels outside the optimal range can lead to decreased viability and increased susceptibility to environmental stressors [8,9,10]. UV radiation, particularly in the UV-C spectrum, is known for its germicidal effects, effectively reducing bacterial viability upon exposure [11,12].

While much research has focused on understanding bacterial survival on solid surfaces or in liquid media, fewer studies have explored the dynamics of bacterial aerosols. Analyzing influencing factors on bacterial survival to airborne scenarios requires careful consideration of additional variables unique to aerosolized environments, including particle size distribution, air flow dynamics, and the presence of atmospheric gases [4,13,14,15].

In this context, the influence of carbon dioxide (CO_2_) concentrations and dust levels on bacterial survival in the air represents a relatively understudied area even though it might play a significant role in climate change [16]. CO_2_, a natural constituent of the atmosphere, can accumulate to elevated levels in indoor environments, particularly in poorly ventilated spaces or areas with high occupant density [17]. It serves not only as a vital component of cellular respiration for many organisms but also plays a role in modulating environmental conditions. In laboratory settings, CO_2_ is often regulated to maintain physiological conditions conducive to bacterial growth and viability in culture media [18,19]. It is routinely supplied to incubators and growth chambers to maintain optimal pH conditions for cell culture and bacterial growth and to enhance microbial proliferation [20]. However, the implications of fluctuating CO_2_ concentrations on airborne bacterial survival remain poorly understood. Investigating the influence of CO_2_ on airborne bacteria is essential not only for understanding their survival dynamics in indoor environments but also for optimizing laboratory conditions to accurately simulate real-world scenarios.

Similarly, dust particles, comprising a complex mixture of organic and inorganic materials, ubiquitous in indoor and outdoor air, have been implicated as carriers for microbial contaminants, including bacteria and viruses [21]. Dust particles can originate from various sources, including skin flakes, textile fibers, pollen, and soil particles, and can harbor a diverse array of microbial species [22,23]. Dust can provide a substrate for microbial attachment and growth, providing nutrients, and protection from environmental stressors, and potentially prolonging the survival of airborne pathogens [21]. Furthermore, the physicochemical properties of dust particles, including size, composition, and surface characteristics, may influence microbial adhesion and survival dynamics [24]. Moreover, dust particles can facilitate the dispersal of bacteria over long distances, contributing to the transmission of infectious agents in both indoor and outdoor environments [25,26]. Despite the recognized role of dust in microbial dissemination, our understanding of its interactions with airborne bacteria and its influence on their survival remains limited.

This work aims to bridge this knowledge gap by investigating the influence of varying concentrations of CO_2_ and dust levels on the survival of two distinct bacterial strains: non-resistant and multi-resistant *Escherichia coli* (*E. coli*). *E. coli* was used for this study as it is a very common laboratory bacterial strain frequently employed in experiments, including experiments on airborne bacteria due to the extensive knowledge available about this species [8,9,10,11,12,27,28]. By systematically altering CO_2_ concentrations and dust levels within an atmospheric simulation chamber with controlled settings, this study aims to provide insights into the complex interplay between environmental factors and bacterial survival in aerosolized form.

## 2. Results and Discussion

### 2.1. Effect of CO_2_ on the Survival Rate

The effect of three different concentrations of CO_2_ (100, 800, 3000 ppm) on the survival rates of both strains (JM109 and JM109-pEC958) was assessed by comparing them to the survival rates of the baseline experiments [29]. These specific concentrations were chosen to encompass a wide range of CO_2_ levels in comparison to the global mean CO_2_ concentration of about 400 ppm, from very low to very high, to ensure that any potential effects on survival rates could be distinctly observed. The results from experiments with CO_2_ set to 100 ppm revealed a reduction in survival rates for both strains, approximately 58% and 46%, respectively. (Figure 1), suggesting that very low CO_2_ concentrations might notably decrease survival rates. CO_2_ plays a crucial role in bacterial metabolism, carbon assimilation, pH regulation, and ecological interactions, with most bacteria requiring a certain level of CO_2_ for survival [30,31,32,33]. The two primary biological processes reliant on CO_2_ for bacteria are the biosynthesis of biomolecules and carbon fixation [34,35]. Although *E. coli* is primarily heterotrophic and typically does not fix carbon from CO_2_, it requires CO_2_ as a carbon source for the biosynthesis of essential biomolecules such as amino acids and nucleotides [36,37,38]. Insufficient CO_2_ concentration (100 ppm) inside the chamber may thus hinder growth and metabolism, explaining the observed reduction in survival rates for both strains.

While CO_2_ is typically essential for bacteria survival at appropriate levels, excessive concentrations can be detrimental due to environmental acidification, disrupting bacterial cellular processes, pH homeostasis, respiration, and membrane integrity [39].

Increasing CO_2_ to 800 ppm had no significant effect on either strain, while increasing it to 3000 ppm significantly reduced the survival rate of JM109 by approximately 24% (Figure 1). Conversely, the JM109-pEC958 was not significantly affected by this CO_2_ concentration, suggesting that JM109-pEC958 can withstand exceptionally high levels of CO_2_ better than its non-resistant counterpart.

The *E. coli* K12 genome contains several CO_2_ utilization-related genes such as those encoding carbonic anhydrases [38]. Carbonic anhydrases are enzymes that catalyze the interconversion between CO_2_ and HCO_3_^−^, crucial for efficient utilization of CO_2_ as a carbon source and intracellular pH regulation [40]. Although pEC958 does not provide additional CO_2_ utilization-related genes, it might contain genes that could potentially regulate the expression of the CO_2_ utilization-related genes on the chromosome, potentially increasing carbonic anhydrases inside the cell and subsequently enhancing resilience to high CO_2_ levels. Additionally, other genes on the plasmid, while not directly related to CO_2_ utilization, might indirectly impact the utilization of CO_2_. Further research is warranted to fully elucidate the mechanism enabling better survival of resistant *E. coli* at exceptionally high CO_2_ levels and how antibiotic resistance genes (ARGs) relate to it.

This finding is significant because it suggests that resistant strains of *E. coli* could outcompete non-resistant ones in high CO_2_ environments, such as poorly ventilated indoor areas, thereby posing a significant health risk.

### 2.2. Effect of Dust on the Survival Rate

Arizona Road Dust comprises relatively large irregular-shaped particles, simulating real-world environmental dust and air pollution scenarios [41]. The experiments were conducted with three different dust concentrations: 250, 500 and 2000 µg m^−3^.

The survival rate of JM109 was significantly reduced, by approximately 26%, at a dust concentration of 250 µg m^−3^ (Figure 2). At this dust concentration, the survival rate of JM109-pEC958 was not yet significantly reduced. At a dust concentration of 500 µg m^−3^ both strains displayed a significant reduction in their survival rates with JM109-pEC958 exhibiting a remaining survival rate of approximately 53%, while that of JM109 was further significantly reduced to 27% compared to the baseline survival rates. The difference in survival rates between both strains was significant at this dust concentration (Figure 3), indicating that the multi-resistant *E. coli* strain can maintain a significantly higher survival rate at high dust concentrations, despite also experiencing a reduction in its own survival rate. This phenomenon was even more pronounced in the experiments with 2000 µg m^−3^ dust, where JM109 exhibited only 13% survival, whereas JM109-pEC958 maintained an approximately 60% survival rate. 

These results suggest that while the survival rates of both strains decrease with increasing dust concentrations, the effect starts at a lower dust concentration for the non-resistant strain. Additionally, the degree of survival rate decrease is more severe for the non-resistant strain compared to its multi-resistant counterpart at 500 and 3000 µg m^−3^.

The presence of dust in ambient air can impose various stressors on bacterial cells, challenging their survival and metabolic activity, thereby explaining the observed reduction in survival rates during our experiments [42]. Stressors may include oxidative stress due to reactive oxygen species (ROS) contained in dust particles, chemical contaminants accumulated in the dust, and physical damage to the bacteria cells through abrasion from the particles’ abrasive surfaces [43,44,45]. Arizona Road Dust used in this study, derived from road surfaces, may contain mineral dust, organic matter, and pollutants that might have undergone photochemical reactions in sunlight, leading to the production of ROS such as superoxide radicals (O_2_^−^), hydroxyl radicals (OH^−^), and hydrogen peroxide (H_2_O_2_) [43,46,47]. These ROS can induce oxidative stress and damage cellular components in the bacterial cells [48]. Furthermore, the dust may contain various chemical contaminants from vehicle emissions, industrial activities, and atmospheric deposition, such as heavy metals and polycyclic aromatic hydrocarbons (PAHs) [44,49]. Exposure to such chemical contaminants can exert toxic effects on cells and disrupt cellular functions. A combination of these stressors in the dust particles might be responsible for survival reduction in the strains during our study.

The fact that JM109-pEC958 was less affected and maintained a higher survival rate than its non-resistant counterpart may be attributed to efflux pumps encoded on the plasmid. Efflux pumps can extrude chemical contaminants from cells, reducing intracellular concentrations of harmful substances [50]. JM109 lacks the genes encoding efflux pumps, whereas pEC958 contains *tetA* and its regulatory gene *tetR*, potentially increasing its ability to remove contaminants from the cells and thus contributing to better survival than JM109 without pEC958 [51,52]. Further research is needed to identify the exact mechanisms provided by pEC958 and potentially other plasmids and resistance genes for better survival at high dust concentrations.

## 3. Material and Methods

### 3.1. Preparation of Bacterial Strains

To assess the disparity in survival between multi-resistant bacteria and their non-resistant counterparts, we utilized the reference strain *E. coli* K12. *E. coli* K12 is a well-studied non-pathogenic *E. coli* strain commonly employed in laboratory settings [53]. For this study, two distinct variants of the strain were utilized: the wild-type *E. coli* K12 and a genetically modified variant containing the plasmid pEC958, which encodes multi-resistance genes. The pEC958 plasmid is derived from the highly resistant *E. coli* variant ST131, which is responsible for numerous infection outbreaks in hospitals [51].

The JM109 High Efficiency Competent Cells (Promega, Madison, WI, USA) served as the non-resistant wild-type strain. To generate the multi-resistant strain, the JM109 strain underwent a transformation by integrating the pEC958 plasmid, following the standard transformation protocol for single-use cells from the competent cells’ manufacturer. The pEC958 plasmid was previously extracted from a clinical sample of *E. coli* ST131 using Qiaprep spin miniprep kit (Qiagen, Hilden, Germany).

### 3.2. Preparation of Bacterial Suspension

To prepare the bacterial suspension for the injection into the chamber, the bacterial strain was cultured overnight on a petri dish containing appropriate media: LB (Merck KgaA, Darmstadt, Germany) media plates for JM109 and LB media plates spiked with 1 mL ampicillin (100 mg mL^−1^, Sigma-Aldrich, Burlington, MA, USA) per liter of agar for JM109-pEC958. Subsequently, the bacterial cells were suspended in 25 mL of LB broth and incubated at 37 °C with continuous shaking until they reached the logarithmic (log) phase of growth. The log phase was determined by achieving an OD_600nm_ (Shimadzu 1900, Columbia, MD, USA) reading of 0.6 [54,55].

Subsequently, 20 mL of the bacterial suspension was centrifuged at 4000 rpm for 10 min, and the resulting pellet was resuspended in 20 mL 0.9% NaCl solution.

### 3.3. Chamber Operations

The experiments were conducted in the Chamber for Aerosol Modelling and Bio-aerosol Research (ChAMBRe) [56,57,58], an atmospheric simulation chamber installed at the National Institute of Nuclear Physics in Genoa, Italy, in collaboration with the Environmental Physics Laboratory of the Physics Department of the University of Genoa. The chamber has a total volume of 2.2 m^3^ and is equipped with several in- and outlets that facilitate aerobiological simulations under controlled conditions. Photos of the setup and a scheme of the experimental procedure can be found in Figure 4 and Appendix A, while a recent detailed description of the facility can be found in [59].

The bacterial cells suspended in 0.9% NaCl were injected into the chamber using the SLAG nebulizer (Sparging Liquid Aerosol Generator, CH Technologies, Westwood, NJ, USA) and an automatic syringe pump (NE-300 Just Infusion™ Syringe Pump, New Era, Farmingdale, NY, USA). The injection process typically lasted 5 min, with a working airflow of 3.5 Lpm. These parameters were automatically controlled by a mass flow controller (MFC, Bronkhorst, Ruurlo, The Netherlands; model F201C-FA) managed via NI LabView^TM^ SCADA (Supervisory Control And Data Acquisition). The syringe rate was set at 0.4 mLpm, resulting in the nebulization of 2 mL of the bacterial suspension. Thereby, a bacterial cell concentration of approximately 10^6^ colony-forming unit (CFU) mL^−1^ inside the ChAMBRe was achieved.

The concentration of the total bacteria inside the ChAMBRe was monitored using a waveband-integrated bio-aerosol sensor (WIBS-NEO; Droplet Measurement Technologies, Longmont, CO, USA), which utilizes fluorescence signals to identify and differentiate bio-aerosol components. A custom procedure for WIBS data reduction, written in Igor PRO 8.0 and previously published by Vernocchi et al. (2023) [59], optimized for the JM109 strain, was employed to retrieve the time series of bacteria concentration throughout the experiment.

A series of “baseline” experiments (i.e., in clean air) was carried out to assess the stress induced by the experimental procedure on the bacteria and to determine, through comparison, the effects of atmospheric components or pollutants on the bacteria viability. “Clean” air was introduced into the chamber through the following procedure [56]: first the chamber was evacuated at least down to 10^−2^ mbar, then, pure N_2_ from a compressed gas cylinder was flushed in, until a pressure of 5 mbar was reached, and then the ambient air re-entered the chamber through an absolute HEPA filter (Kurt J. Lesker, Dresden, Germany, model: PFIHE842, NW25/40 Inlet/Outlet—25/55 SCFM, 99.97% efficient at 0.3 μm) and a zeolite trap (right before the HEPA filter). The ambient conditions during each baseline experiment were set at atmospheric pressure, with CO_2_ concentrations at 400 ppm, temperatures around 20 °C and relative humidity ranging from 60 to 70%. Other gases (O_3_, NO_x_ and SO_x_) and the dust concentration were maintained below the minimum detectable level (MDL) of the respective detector/monitor [56].

Subsequently, the effects of different CO_2_ and dust concentrations were evaluated by maintaining the chamber conditions identical to those during the baseline experiments, except for varying CO_2_ or dust concentrations.

During CO_2_ experiments, the gas concentration was kept constant thanks to an automatic feedback control system [59], and monitored by a non-dispersive carbon monoxide and dioxide analyzer (CO12e, ENVEA, Poissy, France). CO_2_ levels that were tested were 100, 800 and 3000 ppm.

For experiments involving exposure to dust, an OPS (Optical Particle Sizer, model 3330, TSI Inc., Shoreview, MI, USA) was utilized to obtain the dust mass concentration inside the chamber. The dust was generated using a solid particle disperser (Palas, model RBG 1000, Karlsruhe, Germany), which injected Arizona Road Dust (ISO 12103-1, A1) [60] a typical laboratory dust, into the chamber [41,61]. Three different dust concentrations were tested, 250, 500 and 2000 µg m^−3^.

A fan installed at the bottom of the chamber, operating constantly at 5 Hz, ensured homogenization of the concentration and proper mixing within the chamber.

Bacteria were collected via gravitational settling on four petri dishes filled with the appropriate culture medium (LB or LB+ampicillin), positioned at the bottom of the chamber via an automated shelf. The typical duration of an experiment was approximately 5 h. The extracted petri dishes were then incubated overnight at 37 °C, and CFUs were counted the following day.

### 3.4. Determination of Survival Rates

The amount of the bacteria injected into the chamber varied from experiment to experiment. Therefore, a proper correlation procedure was needed to determine the survival rates. First, the concentration of bacterial cells inside the chamber was determined through the following steps:(1)Measurement of OD_600nm_ of the suspended cells in 0.9% NaCl solution to assess the total bacterial concentration injected into the chamber. OD_600nm_ is proportional to the bacteria concentration, thus the equation below was employed to determine the total bacterial concentration:
(1)$${\left(\frac{cells}{ml}\right)}_{estimated}={OD}_{measured}\times 8\cdot 10^{8}\frac{cells}{ml},$$
based on the empirical relation that 1 OD_600nm_ = 8 × 10^8^ cells mL^−1^ for *E. coli* [62]. This method determines the total number of cells, irrespective of viability, providing information solely about the total amount of cells injected into the chamber.(2)Preparation of appropriate serial dilutions of the bacterial suspension in 0.9% NaCl, followed by plating on LB culture medium-filled petri dishes (LB media plates for JM109 and LB media plates spiked with ampicillin for JM109-pEC958). After overnight incubation at 37 °C, CFUs were counted the next morning to estimate the viable fraction of the bacterial suspension in terms of CFU mL^−1^.(3)Calculation of the dead bacteria concentration by subtracting the viable bacterial concentration from step 2 from the total bacterial concentration obtained in step 1.(4)Determination of a correlation factor, Cf, using the following equation:
(2)$$Cf=\frac{\beta }{1+\beta }$$
where:(3)$$\beta =\frac{viable\;bacteria\;concentration}{dead\;bacteria\;concentration}$$(5)Determination of the total airborne bacteria concentration inside the ChAMBRe by the WIBS data analysis as # cm^−3^, considering data at 3 min after the injection’s conclusion to ensure proper mixing within the chamber volume.(6)Calculation of the airborne viable bacteria concentration (# cm^−3^) inside the chamber using the measured airborne bacterial concentrations from step 5 and the correlation factor from step 4:
(4)$${\left(\frac{cells}{{cm}^{3}}\right)}_{viable\;airborne\;bacteria}=Cf\times {\left(\frac{cells}{{cm}^{3}}\right)}_{total\;airborne\;bacteria}$$

This represents the final calculated airborne concentration of viable bacterial cells inside the chamber exposed to different ambient conditions tested during the experiments (see Section 3.3).

Next, the fraction of bacterial cells that survived the 5 h period of each experiment was determined through the following steps:(1)Using the bacteria culturable fraction collected on the four petri dishes inside the chamber (see Section 3.3), determined by CFU visual counting and inserted into the following equation to determine the ratio of the fraction of surviving bacteria:
(5)$$Ratio=\frac{{CFUs}_{petri\;dishes\;inside\;the\;chamber}\;}{{\left(\frac{cells}{{cm}^{3}}\right)}_{viable\;airborne\;bacteria}}$$(2)Calculation of the survival rate (%) of the bacterial cells under specific environmental conditions by comparing the ratios during those experiments to the ratios obtained during baseline experiments:
(6)$$Survival\;rate\;(\%)=\frac{{ratio}_{experiment}}{{ratio}_{baseline}}\times 100$$

The results were presented as percentages. The baseline survival rate is considered 100%, and the survival rates obtained during experiments with varying concentrations of CO_2_ and dust were compared to assess their effect on the survival of the bacterial cells. Several repetitions of each experiment were conducted to increase statistical significance. The variation in the number of repetitions between experiments arose because some experiments reached statistical significance with fewer repetitions, while others required more repetitions to achieve the same level of significance. More details about the baseline determination as well as limitations and challenges of this study can be found in S2.

### 3.5. Statistical Analysis

All statistical analyses and graphical visualizations were conducted using Graphpad Prism (v10.0.2, available at https://www.graphpad.com/, accessed on 2 April 2024). The results were deemed statistically significant when *p* ≤ 0.05.

## 4. Conclusions

To our knowledge, this study represents the first investigation into the impact of varying CO_2_ and dust concentrations on the survival of airborne bacteria, with a comparative analysis between non-resistant and multi-resistant bacterial strains.

The results suggest that multi-resistant *E. coli* strains exhibit greater resilience to high CO_2_ and dust levels compared to their non-resistant counterparts. Notably, at CO_2_ concentrations of 3000 ppm, the survival rate of JM109 was significantly reduced by approximately 24%, whereas the survival rate of JM109-pEC958 remained unaffected. This disparity was even more pronounced during the dust experiments. JM109 experienced a 26% reduction in survival rate already at 250 µg m^−3^, whereas the multi-resistant counterpart showed a significant reduction with the dust concentration at or higher than 500 µg m^−3^. At this concentration, the non-resistant strain exhibited a survival rate of 27%, which further decreased to 20% at 2000 µg m^−3^. In contrast, the multi-resistant strain maintained significantly higher survival rates, approximately 55% at high dust concentrations.

These findings suggest that the JM109-pEC958 strain possesses mechanisms enabling it to endure extreme ambient conditions such as high CO_2_ and dust concentrations better than its non-resistant counterpart. These mechanisms may include regulatory genes or genes encoding efflux pumps. Further research is warranted to elucidate the precise mechanisms underlying this enhanced resilience.

Understanding their mechanisms is crucial for developing effective mitigation strategies to prevent the enrichment and dissemination of antibiotic-resistant bacteria in the air. Additionally, future studies could explore the impact of other environmental factors, such as additional gases and UV light, to determine if similar effects are observed when comparing non-resistant and multi-resistant bacterial strains. Such investigations would provide valuable insights into the adaptation strategies employed by multi-resistant bacteria in response to various environmental stressors, and this study represents the initial step in that direction.

## Figures and Tables

**Figure 1 antibiotics-13-00558-f001:**
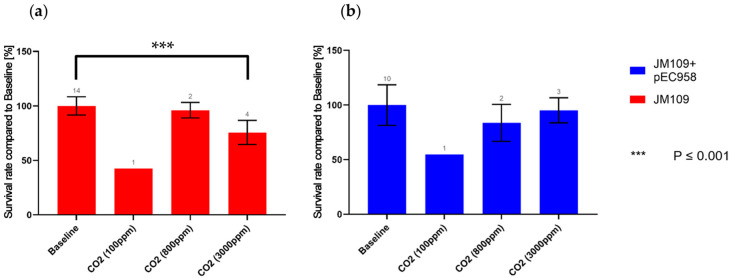
Comparison of the survival rate at different CO_2_ concentrations of (**a**) the wild-type *E. coli* K12 (JM109), (**b**) the modified *E. coli* K12 (JM109-pEC958); the numbers above the histograms are the numbers of experiments.

**Figure 2 antibiotics-13-00558-f002:**
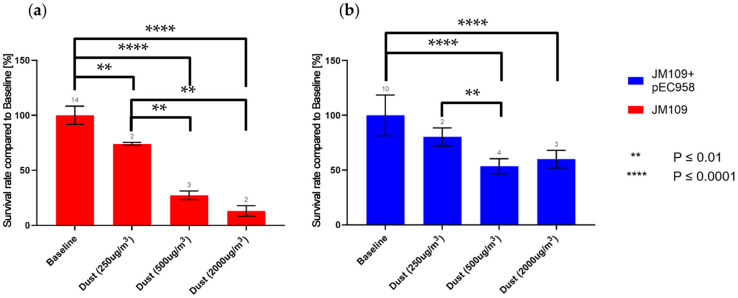
Comparison of the survival rate at different dust concentrations of (**a**) the wild-type *E. coli* K12 (JM109), (**b**) the modified *E. coli* K12 (JM109-pEC958); the numbers above the histograms are the numbers of experiments.

**Figure 3 antibiotics-13-00558-f003:**
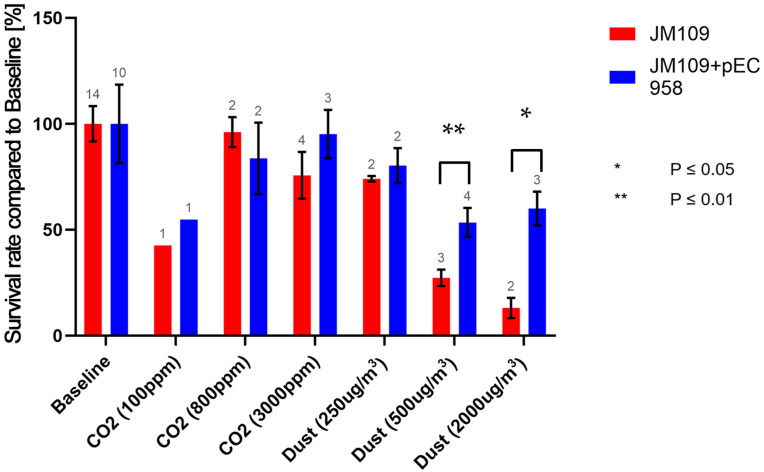
Comparison of the survival rates of both strains (JM109 and JM109-pEC958) at different CO_2_ and dust concentrations; The numbers above the histograms are the number of experiments.

**Figure 4 antibiotics-13-00558-f004:**
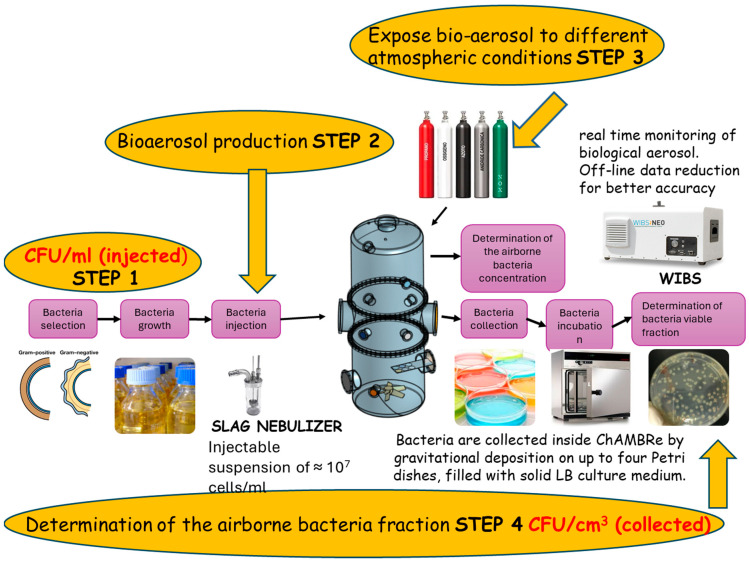
Scheme of experimental procedure.

## Data Availability

The raw data supporting the conclusions of this article will be made available by the authors on request.

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
