# Peer review of "Influence of CO2 and Dust on the Survival of Non-Resistant and Multi-Resistant Airborne E. coli Strains"

_antibiotics, 2024, doi:10.3390/antibiotics13060558_

Round 1
Reviewer 1 Report
Comments and Suggestions for Authors
See attached

Reviewer 2 Report
Comments and Suggestions for Authors
Title: Influence of CO2 and Dust on the Survival of Non-resistant and Multi-resistant Airborne E.coli Strains
The authors show that the drug-resistant strain JM109-pEC958 can tolerate and survive in extreme ambient conditions, such as high CO2 and dust concentrations, better than the respective non-resistant strain. This study provides important insight into how environmental stressors impact drug-resistant bacterial survival. I have a few comments:
1. S2 should be added to the main text.
2. Line 268: ‘stains’ should be ‘strains’.
3. Line 156, 225: Authors should clearly mention why 100ppm, 800ppm, and 3000ppm of CO2 were used. The reference 35 'Govindasamy B. Digesting 400 ppm for global mean CO2 concentration. Curr Sci. 10. Juni 2013;104:1471–2.' used does not provide the information. Authors should provide a relevant reference.
Author Response
Reviewer 2:
- S2 should be added to the main text.
- Thanks for the recommendation. We added S2 to the main text (now: Figure 1).
- Line 268: ‘stains’ should be ‘strains’.
- Thanks for pointing this out. “stains” was corrected to “strains”.
- Line 156, 225: Authors should clearly mention why 100ppm, 800ppm, and 3000ppm of CO2 were used. The reference 35 'Govindasamy B. Digesting 400 ppm for global mean CO2 Curr Sci. 10. Juni 2013;104:1471–2.' used does not provide the information. Authors should provide a relevant reference.
- Thank you for pointing this out. We added another sentence to explain our reasoning for choosing these concentrations: Line 236-8: “These specific concentrations were chosen to encompass a wide range of CO2 levels in comparison to the global mean CO2 concentration of about 400 ppm, from very low to very high, to ensure that any potential effects on survival rates could be distinctly observed.”